# Fast methane diffusion at the interface of two clathrate structures

Umbertoluca Ranieri [1,2], Michael Marek Koza [2], Werner F. Kuhs [3], Stefan Klotz[4], Andrzej Falenty [3], Philippe Gillet[1] & Livia E. Bove [1,4]

Methane hydrates naturally form on Earth and in the interiors of some icy bodies of the Universe, and are also expected to play a paramount role in future energy and environmental technologies. Here we report experimental observation of an extremely fast methane diffusion at the interface of the two most common clathrate hydrate structures, namely clathrate structures I and II. Methane translational diffusion—measured by quasielastic neutron scattering at 0.8 GPa—is faster than that expected in pure supercritical methane at comparable pressure and temperature. This phenomenon could be an effect of strong confinement or of methane aggregation in the form of micro-nanobubbles at the interface of the two structures. Our results could have implications for understanding the replacement kinetics during sI–sII conversion in gas exchange experiments and for establishing the methane mobility in methane hydrates embedded in the cryosphere of large icy bodies in the Universe.

[1] EPSL, ICMP, École polytechnique fédérale de Lausanne (EPFL), Station 3, CH-1015 Lausanne, Switzerland. [2] Institut Laue-Langevin, 71 avenue des Martyrs, CS 20156, 38042 Grenoble cedex 9, France. [3] GZG Abt. Kristallographie, Universität Göttingen, Goldschmidtstrasse 1, 37077 Göttingen, Germany. [4] Institut de Minéralogie, de Physique des Matériaux et de Cosmochimie, Université Pierre et Marie Curie Paris 06, CNRS Unité Mixte de Recherche 7590, Sorbonne Universités, F-75252 Paris, France. Correspondence and requests for materials should be addressed to U.R. (email: ranieriu@ill.fr) or to L.E.B. (email: livia.bove@impmc.upmc.fr)

Gas clathrate hydrates are a general class of compounds composed of water (hosts) molecules and gas (guests) atoms or molecules[1]. Among them, clathrate hydrates of methane are probably the most extensively studied. The natural occurrence of methane hydrate-containing sediments in permafrost areas and ocean shelves, the risk due to their formation in oil and gas pipelines, as well as their potential application as gas transportation media in soft conditions (i.e., close to atmospheric pressure and room temperature) explain the wide interest shown for these materials[1, 2]. Exchanging the guests in natural gas hydrate deposits with $CO_2$ has been suggested as a two-in-one approach of energy recovery and concomitant $CO_2$ mitigation[3]. As they are believed to be the dominant methane-bearing phase in the nebula from which the outer planets and satellites are formed, the properties of methane hydrates are also crucial to models of bodies in the outer solar system[4]. From a physical–chemical point of view, methane hydrates represent prototypical examples of hydrates of hydrophobic guests: the combination of low temperature, high pressure, a weak gas–water interaction between the guest molecules and the host lattice, and the relatively strong hydrogen bonds between host molecules allow for the formation of open crystalline water networks encaging $CH_4$ molecules. The topology of the water cages and the number of gas molecules trapped in these cages critically depend on the specific thermodynamic conditions of formation of the clathrate hydrate and on its formation kinetics[1, 5].

The most common structures formed by clathrate hydrates at relatively moderate pressures are the clathrate structures I and II (noted sI and sII). The unit cell of clathrate sI (space group $Pm\bar{3}n$) contains two small dodecahedral ($5^{12}$) water cages and six bigger tetrakaidecahedral ($5^{12}6^2$) cages. The unit cell of sII (space group $Fd\bar{3}m$) contains 16 $5^{12}$ cages and eight large hexadecahedral ($5^{12}6^4$) cages[1] (Fig. 1). It is well accepted that methane hydrates preferentially crystallise into sI[1]. However, cages characteristic of sII have been transiently detected in the initial stages of the formation of methane hydrates in both experiments[6, 7] and simulations[8–12]. This is not surprising since (i) the difference in free energy between sI and sII is small[13] and (ii) appearance of metastable polymorphs or transient non-equilibrium states is commonly observed during nucleation of hydrates[5, 13–16]. It is noteworthy that sI and sII are topologically incompatible without the intercalation of pentakaidecahedral ($5^{12}6^3$) cages;[8] the interplay between kinetic factors and thermodynamic stability during sI–sII cross-nucleation has been discussed in details[17]. In methane hydrates at room temperature and pressures up to 0.6 GPa, sII has been reported to persistently coexist with sI[18–20]. Therefore, the resulting coexistence of structures in high-pressure samples can be seen as a frozen form on laboratory timescales of the metastable sI–sII polymorphs usually encountered during nucleation of methane hydrates.

Low-temperature translational and rotational excitations, as well as cage-to-cage hopping of $CH_4$ molecules trapped in clathrate sI were previously investigated at ambient and low pressures[21–24]. However, no information is available on the extra-cage diffusivity of the guest molecules in methane hydrates; this information could be highly relevant for the modelling of the subcrustal layers of methane clathrates embedded in the cryosphere of icy planets and large icy satellites[25, 26]. Recently, a study based on molecular dynamics simulations reported diffusion coefficient values in the nanosecond time scale for methane diffusion at grain boundary-like structures of defective clathrates[27].

In this work, we probe the microscopic diffusion of methane in a methane hydrate ($CH_4$–$D_2O$) sample exhibiting coexistence of clathrate sI and sII by quasielastic neutron scattering (QENS) measurements. Coexistence of structures is promoted by applying high pressure. QENS is a well-suited technique to study dynamics on the picosecond time and Å length scales[28]. Spectra of the sI–sII clathrate show a clear quasielastic signal whose analysis reveals a very fast extra-cage translational diffusion of methane molecules on the picosecond time scale. For comparison, we also measure methane hydrates in pure sI clathrate, in pure hexagonal clathrate structure H (space group P6/mmm)[26] and during transformation from sI to structure H (noted sH); the spectra of sI and sH do not exhibit any visible quasielastic signal, and the spectra of sI–sH show a very weak signal, orders of magnitude smaller than the signal from sI–sII.

## Results

**QENS experiments and elastic scattering.** The experiments were performed at the time-of-flight spectrometer IN6 at the Institut Laue-Langevin in Grenoble (France) using a Paris-Edinburgh

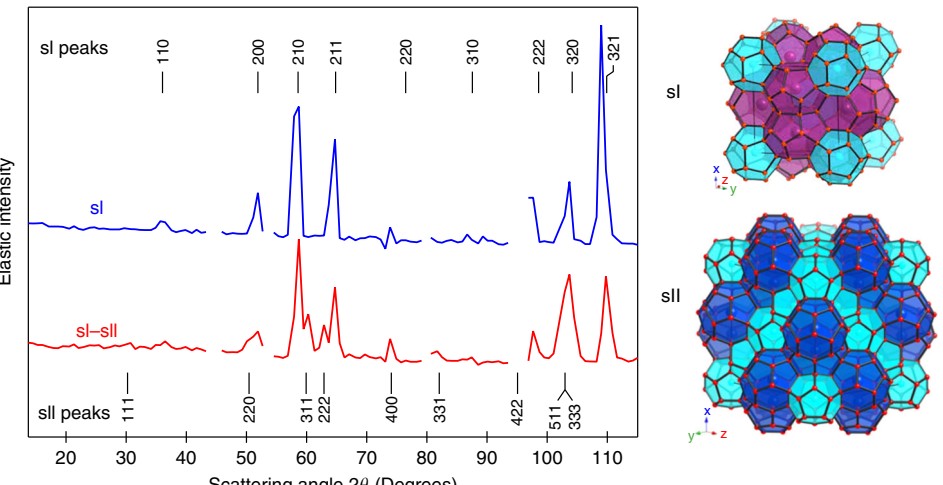

**Fig. 1** Neutron diffraction patterns. Powder diffraction patterns of methane hydrate in pure sI clathrate at 0.4 GPa and 290 K and in the sI–sII clathrate at 0.8 GPa and 282 K. Breaks correspond to noisy detectors and to the strong Bragg peak of alumina from the anvils at 95°. The positions of the Bragg peaks for sI (cell parameter 11.7 Å) and for sII (cell parameter 17.0 Å) are reported. On the right, we present views of the unit cells of sI and sII ($5^{12}$ cages in cyan, $5^{12}6^2$ cages in purple, $5^{12}6^4$ cages in blue)

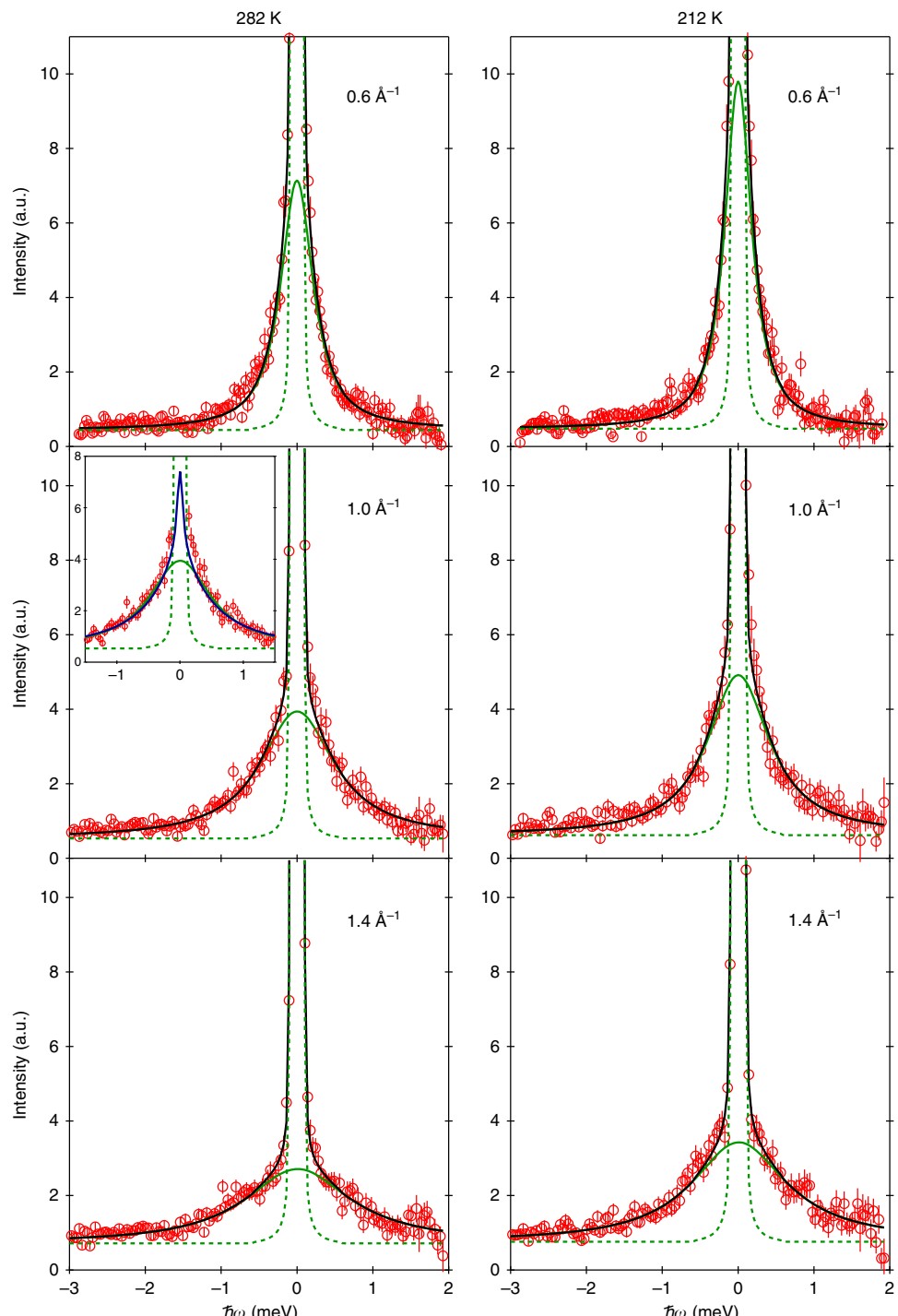

**Fig. 2** Examples of measured QENS spectra. QENS spectra of methane hydrate in the sI–sII clathrate at 0.8 GPa and selected temperature $T$ and momentum transfer $Q$ values. Experimental data (empty circles) are compared to their best fits (black lines). Error bars were calculated by the square root of absolute neutron count combined with the law of propagation of errors. Quasielastic Lorentzian (solid green lines) and elastic (dashed green lines) components are also shown (upshifted by the value of the flat background for clarity). In the inset, a Lorentzian fit is compared to the 2D diffusion fit (blue line) of the same spectrum

press and recently developed ceramics anvils[29]. The wavelength of the incoming neutrons was 5.12 Å, resulting in an instrumental resolution of 0.08 meV. This corresponds to an observation time of ~8 ps. The sample exhibiting coexistence of clathrate sI and sII was prepared according to the following procedure: we compressed methane hydrate (originally in sI) to 0.8 GPa at liquid nitrogen temperature and then warmed it up to 282 K. The neutron powder diffraction pattern of the sample at 0.8 GPa and

282 K is presented in red in Fig. 1. It indicates that the sample contained about half as much sII than sI, on the basis of peak heights. The pattern was obtained directly on IN6 by comparing the intensities of the elastic peaks, at each scattering angle, with those measured on a vanadium standard which gives isotropic elastic scattering. All the Bragg peaks of the sample can be indexed within the space groups of sI and sII. Figure 1 also depicts the diffraction pattern of pure sI clathrate. The diffraction

patterns of pure sH clathrate and of the sample transforming from sI to sH are presented in Supplementary Fig. 1. We recorded QENS spectra of the sI–sII clathrate at the constant temperature $T$ of 282 K and pressure of 0.8 GPa during 6 h. The amount of sII was constant during this time. Then we continuously decreased the temperature to 200 K over 15 h to characterize the $T$ dependence of the probed diffusion at 0.8 GPa. Spectra measured between 282 and 200 K were merged into three groups of 5 h of acquisition time each, corresponding to the following average temperature values: 267, 238 and 212 K. Upon cooling down, the relative amount of sI and sII remained approximately constant. The diffraction patterns recorded at 267, 238 and 212 K are reported in Supplementary Fig. 2. More details about the experiments are given in the Methods section.

**Fitting of the QENS spectra.** Figure 2 depicts typical QENS spectra of the methane hydrate sample exhibiting coexistence of clathrate sI and sII at 0.8 GPa. Examples of spectra of methane hydrate in pure sI clathrate, in pure sH clathrate and in the sI–sH clathrate are shown in Supplementary Fig. 3. Spectra in Fig. 2 show a clear quasielastic signal, i.e., a broadening of the elastic line produced by interactions of the neutrons with diffusing atoms of the sample. Since the incoherent cross-section of hydrogen is almost two orders of magnitude larger than that of other atoms, the measured signal is essentially due to the dynamics of protons in the guest molecules[30, 31]. We first applied the most common model used to fit quasielastic data, i.e., a Lorentzian function (whose half-width-half-maximum is noted $\Gamma$). A delta function was used to fit the elastic line of the spectra, which is due to the contribution of the water network and of the slowly-diffusing or non-diffusing methane molecules trapped in the clathrate structures. Total best fits to the experimental data are presented in Fig. 2 and can be seen to accurately describe the spectra. Based on the integrated areas of the quasielastic and elastic lines (after subtraction of the empty cell measurement), we roughly estimate that about one third of the methane molecules in the sample contribute to the fitted quasielastic signal, at each investigated $T$. More details on this estimation are given in Supplementary Note 1. Since cage occupancies in the newly formed sII clathrate might be lower compared to the starting sI clathrate hydrate, part of the methane molecules in the starting sI clathrate could indeed have been released from the starting sI hydrate into the grain boundary network during transformation from sI to sII and would be available to perform extra-cage translational diffusion. However, a minimum level of occupancies is required to ensure stability of sII and one can estimate that no more than 10% of the methane in the sample could have been released without destabilization of the water matrix. The existence of a fraction of fast diffusing methane molecules as high as one third strongly suggests that an appreciable fraction of water molecules in the sample are in a disordered state. Such disordered regions would form at the front line of the transformation between clathrate sI and sII, and their sizes are most likely far below the typical size of the crystallites (that is a few micrometres[32]). This point is further discussed in Supplementary Note 2. Moreover, the absence of a prominent quasielastic signal in the spectra of the sI–sH methane clathrate hydrate highlights the very particular nature of the interfaces between coexisting sI and sII, compared to the temperature-induced or pressure-induced structural transition taking place at high driving forces between two stable forms of methane hydrates such as sI and sH. The micro-structural properties of sI and sII coexisting assemblies certainly deserve to be further investigated.

**Momentum transfer $Q$ dependence of the QENS signal.** Figure 3 depicts the parameter $\Gamma$ as a function of $Q^2$. The $Q$ dependence of

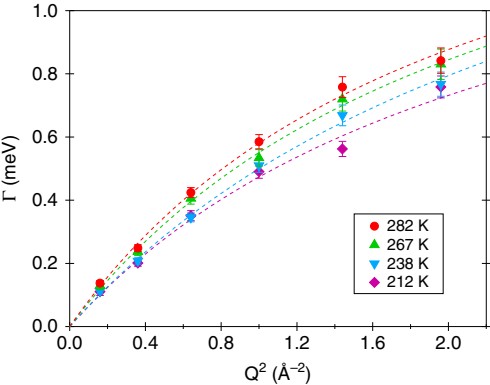

**Fig. 3** Momentum transfer $Q$ dependence of the QENS signal. Half-width-half-maximum $\Gamma$ of the Lorentzian quasielastic component of the fits (Fig. 2) as a function of $Q^2$ at 0.8 GPa and the investigated temperatures. Error bars correspond to one standard deviation. The best fits to the data according to Eq. (1) are shown as dashed lines

$\Gamma$ provides information about the characteristic time and nature of the probed motion. The monotonic increase of $\Gamma$ rules out that the measured quasielastic signal is due to a localised (rotational) dynamics of methane, which would be indicated by a $Q$-independent $\Gamma$. Instead, it clearly highlights that a translational diffusion process is at the origin of the signal[28]. It must be also noted that the rotational quasielastic contribution of methane molecules trapped in clathrate sI is very large (half width above 5 meV) at 150 K[22] and thus only contributes to the background of the spectra here. Similar rotational behaviour can be reasonably expected for $CH_4$ molecules in a clathrate sII, as no indication of inequivalent environments for the guest molecule emerged from the low-temperature rotational spectra of sI methane clathrate[23] (although methane occupies the two types of cages of sI). As can be seen in Fig. 3, $\Gamma$ extrapolates to 0 for $Q \to 0$. Hence, the measured quasielastic signal is not associated with an intra-cage diffusive motion of $CH_4$ molecules, since for a particle restricted to move in a confined geometry $\Gamma$ shows[30, 31] a plateau at small $Q$. For example, for a particle moving within a sphere of radius $R$, $\Gamma$ shows[33] a plateau for $Q < \pi/R$. The $Q$ dependence of $\Gamma$ is best approximated within the random jump diffusion model of Singwi and Sjolander by:

$$\Gamma(Q) = \frac{\hbar DQ^2}{1 + DQ^2\tau}, \tag{1}$$

with $D$ representing the isotropic translational diffusion coefficient and $\tau$ the mean residence time between jumps[28]. The corresponding formula for a continuous free translational diffusion process would be $\Gamma(Q) = \hbar DQ^2$. Fits of $\Gamma(Q^2)$ according to Eq. (1) are presented in Fig. 3; the values obtained for $D$ and $\tau$ are reported in green in Fig. 4. The translational diffusion coefficient turns out to be of the order of $10^{-4}$ cm$^2$ s$^{-1}$ and its temperature dependence is rather weak (25% over the investigated $T$ range). An Arrhenius fit of $D$ provides an activation energy of $0.48 \pm 0.11$ kcal mol$^{-1}$. This value is small compared to the activation energies reported in literature for the cage-to-cage hopping of $CH_4$ in sI clathrates (for example, 12.4 kcal mol$^{-1}$ in ref. [21]) and points at van der Waals interactions as main rate-limiting interactions for the observed methane diffusion. The parameter $\tau$ is a fraction of picosecond and does not show any temperature dependence within the error bars over the investigated $T$ range.

**2D diffusion model.** The choice of a Lorentzian fit function for the quasielastic signal implicitly assumes that the probed motion is three-dimensional (3D)[28]. An other possibility is that the

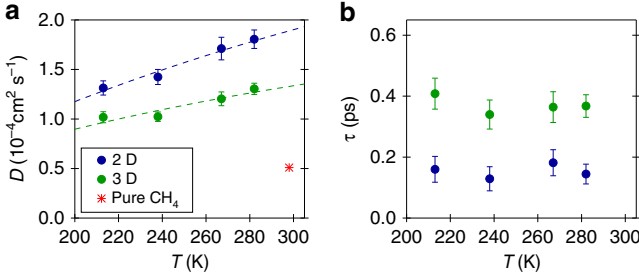

**Fig. 4** Translational diffusion coefficient $D$ and mean residence time $\tau$. Temperature dependencies of $D$ (**a**) and $\tau$ (**b**) for methane hydrate in the sI–sII clathrate at 0.8 GPa, as obtained in the 3D and 2D diffusion models employed in this work. Error bars correspond to one standard deviation. Arrhenius fits of $D$ are shown as dashed lines. Estimated value for $D$ in pure methane from literature[40, 41] is also reported. Legend of **a** also applies to **b**

methane diffuses essentially bi-dimensionally on the length scale probed by the instrument, if the grain boundary network or the intercalated disordered regions between crystals of sI and sII are sufficiently thin. In such a case the fit function for the quasielastic signal is no longer Lorentzian and has a logarithmic singularity at $\omega = 0$[34] (see Supplementary Note 3 for its expression). Nevertheless, the singularity is suppressed by the convolution with the instrumental resolution and the convoluted fit function differs from the convoluted Lorentzian only near $\omega = 0$ where it is more peaked[35]. The inset of Fig. 2 depicts an example of fit using this 2D diffusion fit function and compares it to the Lorentzian fit of the same QENS spectrum. The two fits are actually indistinguishable outside the instrumental resolution-dominated region close to $\omega = 0$ and this is true for all other measured spectra. Therefore it is not possible to unequivocally establish if bulk or planar diffusion is taking place. The values for the translational diffusion coefficient and the mean residence time obtained in the 2D diffusion model are reported in blue in Fig. 4. The activation energy ($0.57 \pm 0.12$ kcal mol$^{-1}$) is comparable to that obtained in the 3D diffusion model. More details about the data analysis are given in the Methods section.

## Discussion

The methane diffusion probed in the present study is much faster than that reported in the literature for the cage-to-cage hopping of $CH_4$ molecules through clathrate sI. Cage-to-cage hopping is a rare event that requires distortion of the host network[36] and the associated diffusion coefficient is of the order of $10^{-11}$ to $10^{-12}$ cm$^2$ s$^{-1}$ at 250 K, as revealed by experimental[21] and computational[24] studies. Similar conclusions have been reported for the cage-to-cage hopping of other guest molecules[37, 38], including molecules forming clathrate sII[37]. It is also interesting to compare the present results to the translational diffusion coefficients of $CH_4$ in bulk water–methane mixtures and bulk pure methane. At 0.02 GPa and 273 K, the diffusion coefficient of methane in water was found to be $0.16 \times 10^{-4}$ cm$^2$ s$^{-1}$, i.e., an order of magnitude smaller than those measured here[39]. This value was obtained for the moderate methane-saturated concentration that is possible at low pressures[39]. In pure methane the experimental diffusion coefficient is $2.08 \times 10^{-4}$ cm$^2$ s$^{-1}$ at 0.164 GPa and 298 K[40]. Its temperature dependence is rather strong, with an activation energy of ~1.0 kcal mol$^{-1}$ between 223 and 323 K at 0.15 GPa. The diffusion coefficient at 0.8 GPa can be estimated based on the assumption that its product with the shear viscosity is constant along isotherms (Stokes–Einstein relation). The pressure dependence of the viscosity in methane at 298 K is known[41] and one gets a value of $0.5 \times 10^{-4}$ cm$^2$ s$^{-1}$ at 0.8 GPa and 298 K. This value is reported in

Fig. 4 and is a factor of 2–3 smaller than our results extrapolated at the same $T$. Based on the same assumption, it is possible to estimate that pure methane at about 0.2–0.3 GPa should show a diffusion coefficient comparable to that measured here.

To summarise, we observed a remarkably fast mobility of methane molecules at the interface of two clathrate structures (I and II) and measured the associated translational diffusion coefficient $D$ at 0.8 GPa and temperatures between 212 and 282 K. The obtained coefficients are 7–8 orders of magnitude higher than those reported in literature for cage-to-cage hopping of methane through clathrate sI at low pressure, one order of magnitude higher than that of methane dissolved in water at low pressure and a factor of 2–3 higher than that expected for pure bulk supercritical methane at comparable pressure and temperature. The activation energy (of about 0.5 kcal mol$^{-1}$) is a factor of two smaller than that of pure methane at 0.15 GPa and more than one order of magnitude smaller than that of the hydrogen bond in the water network and of the cage-to-cage hopping process as reported in literature[21]. This fast mobility involves a sizable fraction of the methane in the sample (approximately one third, as rough estimation), does not induce destabilization of the clathrate structures and is probably observable for times much longer than the duration of our experiment (~21 h).

We infer that the rapidity of the methane diffusion probed here could be an effect of confinement in the extensive grain boundary network[32] or intercalated disordered regions between crystals of clathrate sI and sII. Similar behaviour was already reported in literature. For example, the diffusivity of $CH_4$ is only $4 \times 10^{-11}$ cm$^2$ s$^{-1}$ in zeolite 4A[42], ~$10^{-4}$ cm$^2$ s$^{-1}$ in silicalite[43] and is predicted to be of the same order of magnitude as that of the gas phase ($10^{-1}$ cm$^2$ s$^{-1}$) in infinitely long single-walled carbon nanotubes[44]. Alternatively, the observed fast diffusion could also well be explained by the spontaneous formation of micro-scale to nano-scale gas bubbles from a supersaturated water–methane mixture. Micro-nanobubbles formation was suggested to occur after decomposition of hydrates in different works[45–47]. The diffusion properties of methane inside these bubbles can be considerably different from the bulk fluid and a first study of $CH_4$-mobility in nanobubbles suggested indeed an enhanced diffusion[48]. Further investigation including large-scale molecular dynamics simulations of the guest diffusivity at the structures interface are needed to shed light on these points.

In the context of energy recovery from natural gas hydrate deposits with $CO_2$ injections, gas replacement rates are key parameters to assess its technological viability. Earlier experimental evidences underlined greatly enhanced replacement rates during sI–sII conversion[49, 50] in comparison to the case of iso-structural sI–sI replacement[51]. If extended to moderate pressures, our results might provide an explanation for that. Likewise, our results should be taken into account in the modelling of methane clathrates layers existing at depth in the interiors of large icy bodies in both solar and extra-solar systems[25, 26], for which the steady-states depend on the diffusion timescales as compared to the formation and dissociation rates. As an example, the observed fast mobility of methane could be relevant to understand the phenomenon of methane release into the atmosphere of Titan, which is likely to originate from methane clathrates embedded in its crust and mantle[25, 26].

## Methods

**Sample production**. The procedure followed to prepare the $CH_4$–$D_2O$ methane hydrate sample was described in refs. [21, 52]. It basically consists in keeping $D_2O$ ice under an atmosphere of 6 MPa of $CH_4$ gas at a temperature close to the melting during 4 weeks. The starting deuterated ice was a powder of ice Ih of spherical shape (typical diameter of several tens of micrometres[21]) previously produced by a shock-freezing method through spraying liquid $D_2O$ (99.9% deuterated) into liquid nitrogen. The spraying was done in a glove box under dry nitrogen atmosphere to

avoid contamination with atmospheric water. The quality of the prepared methane hydrate sample was checked by X-ray diffraction. We found that the sample was in clathrate sI with a negligible amount of water ice impurity (below 2%). The size of the crystallites is typically a few micrometres[32]. Typical methane occupation is 86% for the small cages and 99% for the large cages[52].

**Experimental details.** The QENS experiments were carried out using the VX5 Paris-Edinburgh press. The procedure of loading the methane hydrate sample in the clamp module[53] of the press was done under liquid nitrogen. The sample was first compacted to a spherical pellet (of ~40 mm$^3$) using a dedicated press operating under liquid nitrogen. The pellet was subsequently loaded into a precooled type-25 copper-beryllium encapsulating gasket and the sample-gasket assembly was placed in an aluminium ring between precooled ceramics anvils. We used recently developed zirconia-toughened alumina ceramics anvils which are highly transparent to neutrons. Their performances are described in ref. [29].

To prepare the sample exhibiting coexistence of clathrate sI and sII, the gasket was sealed by applying a load of 100 kN on the anvils under liquid nitrogen. This corresponds to a pressure of about $(0.8 \pm 0.1)$ GPa in the sample, on the basis of our calibration of the used anvils. The assembled clamp was then warmed up from liquid nitrogen temperature to room temperature out of the beam before insertion (~12 h later) in the Paris-Edinburgh press. During the experiment, temperature was decreased by cooling down the whole Paris-Edinburgh press in a liquid nitrogen cryostat. It is known that the cooling of samples in such a pressure cell is approximately isochoric and this leads to a small pressure drop (typically below 5% for a change in temperature between 282 and 200 K). The measured Bragg peaks did not shift with temperature within the angular resolution of the instrument.

During a different sample loading, the gasket was sealed by applying a smaller load (50 kN), corresponding to a sample pressure of 0.4 GPa. After being warmed up to 290 K, this sample was still in pure structure I (see Fig. 1 for the diffraction pattern). We compressed this sample isothermally at 290 K and observed transformation to clathrate sH, in agreement with previous studies[20, 26].

During another sample loading, the gasket was sealed by applying a higher load (120 kN), corresponding to a sample pressure of 1.0 GPa. After being warmed up to 295 K, this sample was found to contain a mixture of structure I and structure H. The relative amount of structure H was found to slowly increase over time and the transformation was completed within ~12 h.

The instrumental energy resolution was estimated by measuring a sphere of vanadium of the same size as the sample, which was loaded into the gasket and the Paris-Edinburgh press in the same set-up as the sample at ambient pressure and ambient temperature.

**Data analysis details.** The scattering angles $2\theta$ covered by the detectors of IN6 are in the range 10°–115°. Spectra measured by several detectors were grouped together into constant-Q spectra with 0.2 Å$^{-1}$ steps, from 0.4 to 1.8 Å$^{-1}$. For the data analysis, we did not consider the two highest Q values (1.6 and 1.8 Å$^{-1}$) for which competition of the quasielastic signal with the flat background gives rise to large uncertainties for the free-fitting parameters. Six free-fitting parameters were used in the data fitting with the 3D diffusion model: intensities and half-width-half-maximum of the Lorentzian and delta functions, flat background and zero-shift of the energy-transfer axis. Six free-fitting parameters were used as well in the data fitting with the 2D diffusion model, $D_{2D}(Q)$ substituting the half-width-half-maximum of the Lorentzian. Stokes/anti-Stokes detailed balance and convolution with the instrumental energy resolution function were taken into account. Multiple scattering contribution to the spectra was neglected as the estimated sample transmission is about 89% of the incident beam.

**Data availability.** Raw data were generated at the Institut Laue-Langevin large-scale facility. Derived data supporting the findings of this study are available from the corresponding authors upon request.

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

## Acknowledgements

This work was supported by the Swiss National Science Foundation through FNS Grant 200021–149847, and by the French state funds managed by ANR within the Blanc International programme PACS (reference ANR-13-IS04-0006-01) and the Investissements d'Avenir programme (reference ANR-11-IDEX-0004-02) and more specifically within the framework of the Cluster of Excellence MATériaux Interfaces Surfaces Environnement (MATISSE) led by Sorbonne Universités. We acknowledge the Institut Laue-Langevin for provision of beam time through LTP 6-6, and Claude Payre and James Maurice for technical assistance during the experiments. We thank José Teixeira (LLB) and Robert Pick (IMPMC) for a critical reading of the manuscript.

## Author contributions

U.R, W.F.K. and A.F. prepared the sample. U.R, M.M.K., W.F.K., S.K. and L.E.B. performed the experiments. U.R, M.M.K. and L.E.B. analysed the data. All authors discussed the results and contributed to writing the manuscript.

## Additional information

**Competing interests:** The authors declare no competing financial interests.

