## [Peer Review file · Nature Communications]

Reviewers' comments:

Reviewer #1 (Remarks to the Author):

The authors performed quasielastic neutron scattering measurements of the diffusion coefficient of CH₄ in clathrate. They found the diffusion of CH₄ is faster than the diffusion of bulk CH₄. They interpret the results as indications of fast diffusion of CH₄ at the interface of the two common clathrate structures.

These measurements and data analysis are routine work nowadays. I don't have any problems with that.

But my main problem is: they found some faster diffusion mode of CH₄ at the interfaces, please excuse my language and I'm not trying to be cynical, so what? There is large room at the grain boundaries. Of course the surface diffusion would be faster. That's it. It is good to confirm that. I would recommend the paper to be published on an archive journal. But I really don't see the scientific value at all.

On the technical side, how can the authors confirm the mobile methane are indeed near the interfaces?

Reviewer #2 (Remarks to the Author):

This manuscript reports what is a potentially rather interesting experimental study of fast diffusive motion of methane in a polycrystalline clathrate hydrate material consisting of two structures, namely sI and sII. The major claim in the work, the identification of a fraction of the methane molecules that apparently exhibit translational diffusion that is faster than that expected in a dense supercritical methane fluid, would be of great interest and broad implication if true. It would certainly influence thinking in the field if it can be confirmed. The identification of this behavior relies essentially exclusively on the fitting of spectra obtained from quasielastic neutron scattering (QENS) experiments performed on samples at high pressure. Unfortunately, I am not convinced of the validity of the findings. This is a result both of inconsistencies in their findings as well as a lack of secondary measurements supporting this somewhat surprising claim. Hence, I believe consideration of publication of this study remains premature, as further work is needed to fully validate the findings and their reproducibility. Below I have provided further details of my concerns.

1) Unless I have misunderstood something in the paper, it is unclear where the highly mobile methane has come from. Let me review some of the key data from the study.

i) The study starts with a sample of sI methane hydrate in which the cage occupancy is very high. The sample has negligible excess water (as ice) and (presumably) methane. It is this sample that is compressed to yield the polycrystalline sI and sII material.

ii) The authors report that the compressed material contains roughly 2/3 sI and 1/3 sII. We might then assume that methane from the starting sample is correspondingly disturbed.

iii) Yet, the authors also report that the highly mobile methane in their sample (which they describe as being "extra-cage") comprises roughly 1/3 of the total methane.

But where did the methane exhibiting "fast extra-cage translational diffusion" come from? If the sI and sII hydrate structures have maintained a high level of occupancy, and if there is no significant amount of water/ice presence (from decomposed hydrate), this remains a crucial but unresolved question.

2) It remains unclear the possible nature of the polycrystalline material obtained, and hence the nature of the highly mobile methane within this material.

- a) There is a lack of any data from secondary measurements to support, for example, the existence of a significant fraction of methane that is "extra-cage".
- b) There is also limited information on the nature of the material that results from compression. The only other technique (beyond QENS) that was employed to characterize the sample is the powder X-ray diffraction patterns reported in Fig. 1. The presence of the appropriate Bragg peaks do indicate that both sI and sII hydrate structures are present, but otherwise we have no information on the possible nature of the material (for example: Crystallites of the initial sI hydrate are reported to be a few microns in size - does this remain the case? What might be the nature of the grain boundaries?). If it were somehow possible, microscopy images could also be very insightful.

Reviewer #3 (Remarks to the Author):

The authors report enhanced methane diffusion in mixed sI and sII hydrates based on QENS measurement. The measured methane diffusivity ($\sim 10^{-4}$ cm²/s) is 8 orders of magnitude higher than that observed in pure sI hydrates ($\sim 10^{-12}$ cm²/s), and even a factor of 3 to 4 higher than that for pure bulk methane at comparable temperature and pressure. The results is surprising and unexpected and may stimulate great interest in related studies. This work is recommended for publication after minor revision addressing the comments below. Further review is not needed.

1. This is probably the first work that utilize QENS for determining the methane translational diffusion in clathrate hydrates. While the authors have presented different data analysis to ensure that the measured data correspond to methane translational diffusion, it is somehow strange that a measurement of methane diffusion in pure sI hydrates is not provided. Such data have been obtained by other experimental techniques and can serve as a strong support for the use of QENS for the same purpose. Please provide some reasoning if such a straightforward comparison is not possible.
2. The authors state that the measured diffusivity corresponds to the migration of methane at the interface of two clathrate structures, grain boundary network of sI and sII. This is suprisingly similar to the recent work of Lo et al. (10.1021/jp310972b) where the authors reported enhanced methane diffusion at grain boundary like structures based on molecular dynamics simulations. The authors should reference this work as it is quite relevant.
3. The enhanced methane diffusion in mixed sI-sII seems to imply an increased amount of defects in the crystalline phase compared to pure sI hydrates. However, it has been shown that there can be good interfacing between the two structures through the 5₁₂6₃ cages (see the work of Vatamanu and Peter G. Kusalik (10.1021/ja066515t)). The authors might consider SEM measurements for the morphology for their system (for example, as in the work of Falenty et al. (10.1021/jp310972b)) to better understand their system. If such measurements cannot be done by the authors, it is suggested that they comment on the possible interface structures between the sI and sII and why similar interfaces do not appear in pure polycrystalline sI.

Response to comments by reviewer #1:

The authors performed quasielastic neutron scattering measurements of the diffusion coefficient of CH₄ in clathrate. They found the diffusion of CH₄ is faster than the diffusion of bulk CH₄. They interpret the results as indications of fast diffusion of CH₄ at the interface of the two common clathrate structures.

These measurements and data analysis are routine work nowadays. I don't have any problems with that.

We thank the reviewer for having considered our manuscript and for outlining his/her point of view. The reviewer claims that our measurements and data analysis are "routine work". We feel the reviewer disregarded the technical challenge behind such kind of study and we try to defend our point of view below.

QENS studies at GPa pressures have become possible only recently thanks to the use of new technology developed in our group [L.E. Bove, J. Philippe, S. Klotz, patent 1358938 (2013); S. Klotz et al., *Appl. Phys. Lett.* 2013, 103, 193504] and, to our knowledge, the only QENS study existing in literature beyond the kbar range concerns our measurements on liquid water diffusion [L. E. Bove et al., *Phys. Rev. Lett.* 2013, 185901, 1-5]. We think that QENS measurements under pressures in the GPa range are definitely not routine work. Furthermore, the QENS measurements on methane clathrate hydrates presented here introduced a supplementary technical challenge as the loading of the sample into the high-pressure cell had to be performed under liquid nitrogen (to prevent destabilisation of the sample).

Finally, as underlined by reviewer #3, this is the first work directly measuring methane diffusion in clathrate hydrates and is further complicated by the small fraction of hydrogenated molecules in the sample (one CH₄ molecule every six D₂O molecules). Methane inter- and extra-cage diffusion has never been measured in such pressure-temperature domain which also allowed us stabilising the sI and sII coexistence.

But my main problem is: they found some faster diffusion mode of CH₄ at the interfaces, please excuse my language and I'm not trying to be cynical, so what? There is large room at the grain boundaries. Of course the surface diffusion would be faster. That's it. It is good to confirm that. I would recommend the paper to be published on an archive journal. But I really don't see the scientific value at all.

We certainly disagree with the reviewer's concluding statement – our results are of great interest for chemical physicists working on gas hydrates; this was clearly stated by both reviewers #2 and #3. Yet, we are thankful as his/her remark gives us the opportunity to better explain our unusual findings. Indeed, the mobility of extra-cage methane observed here is very **significantly faster than what could be expected from simple grain boundaries** between gas hydrate crystallites. This will be explained in the following in more details.

Of course it is generally agreed that grain boundaries (GB) is a location with higher mobility than in the bulk crystal. This was indeed suggested more than fifteen years ago for CH₄ diffusing in ice [S. Takeya et al., *J. Phys. Chem. A* 2001, 105, 9756-9759] and has been recently confirmed by a molecular dynamics study of sI-sI grain boundaries in methane hydrate [H. Lo et al. 2017 *J. Phys. Chem. C* 121: 8280-8289]. The works of Takeya et al. and Lo et al. show the expected enhanced mobility of methane in grain boundaries by 3-4 orders of magnitude when compared to diffusion in the bulk. Yet the values reported by Takeya et al. and Lo et al. are still **3-4 orders of magnitude smaller than what we observe in our QENS experiments!** We consider our findings extraordinarily remarkable, completely unexplained by anything in literature and thus well worth a publication in Nature Communications.

Let us summarize the relevant orders of magnitude: In our coexisting clathrate sI-sII mixture we measured methane diffusion coefficients of about 10⁻⁴ cm²/s, that is 7-8 orders of magnitude higher than that reported in literature for bulk sI methane clathrate hydrate at low pressure [refs 21 and 24 of the manuscript] and 3-4 orders of magnitude higher than in ordinary grain boundaries of methane hydrate [S. Takeya et al., *J. Phys. Chem. A* 2001, 105, 9756-9759; H. Lo et al. 2017 *J. Phys. Chem. C* 121: 8280-8289]. Even more surprising, the extra-cage methane diffusion we observed in the sI-sII clathrate is also one order of magnitude faster than that of methane in water at low pressure [ref 39 of the manuscript] and **a factor of 2-3 faster than that of pure supercritical methane** at the same pressure and comparable temperature

[refs 40 and 41 of the manuscript]. We would like to emphasize that we are not describing here a limited enhancement due to trivial confinement, but a phenomenon which is **intimately linked to the particular nature of sI-sII interfaces**; as we will show below (and in the revised version of the manuscript and of the Supplementary Information) no comparable signal is observed in other types of grain boundaries (such as sI-sH). It is exactly the very unusual and peculiar behaviour of the coexisting sI-sII mixture that makes our findings so interesting.

In addition, our results provide one of the few experimental examples of the effects of confinement on diffusion of molecular fluids, which are found – mainly by simulations results – in very disparate systems ranging from water in nanotubes [J. R. Bordin et al., *J. Phys. Chem. C* 2014, 118, 9497-9506; A. Barati Farimani et al., *J. Phys. Chem. B* 2011, 115, 12145-12149] to fluids in zeolites [B. Vujić and A. P. Lyubartsev, *Modelling Simul. Mater. Sci. Eng.* 24 (2016), 045002]. Finally, our results can have important implications in the context of CO₂-CH₄ gas exchange experiments and of the modelling of methane clathrate layers existing in the interiors of large icy bodies in the Universe. Nowadays such modelling is based on the very slow diffusion coefficients reported in literature for pure sI methane clathrate hydrate [see for example O. Mousis et al., *Space Sci Rev* (2013) 174, 213-250].

On the technical side, how can the authors confirm the mobile methane are indeed near the interfaces?

The reviewer asks a relevant question. Our interpretation results from a number of experimental evidences. First, the analysis of the quasielastic signal (pag.6-9 of the manuscript) indicates that the probed motion is associated with a translational diffusion (and not an intra-cage motion): this is straightforward as the width of the QENS signal is Q dependent and its low-Q behaviour does not saturate to a finite value, as in the case of dynamics in strong confinement such as intra-cage motion.

Second, the diffraction pattern of the sample, and its invariance under temperature and time (now reported in the Supplementary Information), demonstrates that the crystalline portion of the sample is very persistent and remains in a steady state.

Third, the timescale and the activation energy of the observed diffusive motion are incompatible with both the diffusion of pure methane or of methane dissolved in water.

The only interpretation compatible with these three observations is that the motion takes place outside the cages of the crystalline bulk material, most likely at interfaces between the crystalline parts.

It should be also noted that the small value for the activation energy obtained in our analysis (0.5 kcal mol⁻¹) consistently points at van der Waals interactions as main rate-limiting molecular interaction for the observed methane diffusion. Cage-to-cage hopping of CH₄ in sI clathrates has an activation energy which is more than 1 order of magnitude bigger than our result, as it requires breaking at least one hydrogen bond of the water network [refs 21 and 36 of the manuscript]. In the new version of the manuscript, this point is now commented (in violet) a pag.8 and 10.

Response to comments by reviewer #2:

This manuscript reports what is a potentially rather interesting experimental study of fast diffusive motion of methane in a polycrystalline clathrate hydrate material consisting of two structures, namely sI and sII. The major claim in the work, the identification of a fraction of the methane molecules that apparently exhibit translational diffusion that is faster than that expected in a dense supercritical methane fluid, would be of great interest and broad implication if true. It would certainly influence thinking in the field if it can be confirmed. The identification of this behavior relies essentially exclusively on the fitting of spectra obtained from quasielastic neutron scattering (QENS) experiments performed on samples at high pressure. Unfortunately, I am not convinced of the validity of the findings. This is a result both of inconsistencies in their findings as well as a lack of secondary measurements supporting this somewhat surprising claim. Hence, I believe consideration of publication of this study remains premature, as further work is needed to fully validate the findings and their reproducibility. Below I have provided further details of my concerns.

We would like to thank the reviewer for very carefully reading of our paper and for the constructive criticism. We address all well-taken concerns below and apologize for (apparently) having given only insufficient explanations of our findings in the first version of the manuscript. We think that the clarifications reported below and cast into the revised version of the manuscript alleviate all open issues and make this paper fully suitable for publication in Nature Communications. In the revised version of the manuscript changes addressing the concerns of reviewer #2 are in blue.

1) Unless I have misunderstood something in the paper, it is unclear where the highly mobile methane has come from. Let me review some of the key data from the study.

i) The study starts with a sample of sI methane hydrate in which the cage occupancy is very high. The sample has negligible excess water (as ice) and (presumably) methane. It is this sample that is compressed to yield the polycrystalline sI and sII material.

Remark i) is correct, but there is no excess methane in the starting sample. The starting sI methane hydrate sample was formed from a well-characterised powder of ice Ih and methane gas at high pressure (details of the preparation method are given in refs 21 and 52 of the manuscript). It has been carefully characterized by Rietveld refinements; only a very small amount of relict ice was found and the cage occupancy in large cages was 99% while small cages were filled to around 86%. Once the transformation from ice Ih and CH₄ gas to sI methane hydrate was completed (after 4 weeks of reaction), the sample was recovered and stored. Thus, all methane in the starting sample was trapped in the cages of the clathrate structure I.

ii) The authors report that the compressed material contains roughly 2/3 sI and 1/3 sII. We might then assume that methane from the starting sample is correspondingly disturbed.

Remark ii) is correct, yet with an important disclaimer: the *crystalline* part of the compressed material contains roughly 2/3 sI and 1/3 sII.

iii) Yet, the authors also report that the highly mobile methane in their sample (which they describe as being “extra-cage”) comprises roughly 1/3 of the total methane.

Remark iii) is correct: the highly mobile methane in the sample comprises roughly 1/3 of the total methane.

But where did the methane exhibiting “fast extra-cage translational diffusion” come from? If the sI and sII hydrate structures have maintained a high level of occupancy, and if there is no significant amount of water/ice presence (from decomposed hydrate), this remains a crucial but unresolved question.

The existence of considerable amounts of extra-cage methane in the compressed sample may seem surprising at first sight. Yet, we can offer explanations for this unusual finding which are developed in the following. As apparently our arguments were not presented with sufficient clarity in the first version of the manuscript, we also provide additional material and address the issue of extra-cage methane not only in our reply but also in the revised version of the manuscript in a more explicit manner.

The starting sI methane clathrate hydrate sample is in a stable and equilibrated phase, where all water molecules are part of the crystalline structure and all methane molecules are trapped in the cages of the structure (as indicated in our answer to point 1i). On the contrary, the compressed sample shows coexistence of stable structure I and metastable structure II; such coexistence in near equilibrium is likely characterised by a continuous dynamical rearrangement of water and methane molecules at phase boundaries. This is what we mean by “extra-cage methane”. Our study indicates that the rearrangement involves a fraction of the total methane as high as 1/3. Undoubtedly, the prolonged coexistence of sI and sII originates from the small difference in free energy between the two clathrate structures [ref 13 of the manuscript], so that the driving force for the transformation is very low and the total amount and relative proportions of the two clathrate structures appeared constant throughout the duration of our experiment (21 hours). We added to the Supplementary Information the diffraction patterns collected on the sI–sII hydrate at all investigated temperatures and over time (Supplementary Fig.2). They show that the obtained

polycrystalline material is indeed a mixture of clathrate sI and sII whose total amount *and* relative proportions do not change over temperature and time.

During coexistence of clathrates sI and sII, the two structures have been suggested to develop intercalated micrometer-sized thin layers (see microscopy image below from ref 6 of the manuscript). The disordered regions where methane is able to diffuse would form in between them and their liquid-like contribution to the diffraction patterns would be hardly detectable compared to a bulk amorphous or liquid. It is well conceivable that a water-methane mixture with its high concentration of mobile methane would not or only badly crystallize in the temperature range explored and thus would remain hardly discernible in the low-quality powder patterns obtained on the QENS instrument.

Figure 1: Optical microscopy image of methane hydrates during transformation of sII into sI (from ref 6 of the manuscript). The width of the photographed area is 420 micrometers. The striations observed can be interpreted as layered phase segregations with inherent contrasts in optical density between the involved phases.

In addition, another concurrent source of extra-cage methane is also possible, as indicated in the following. The cage occupancy was 99% in large cages and 86% in the small cages in the starting sI clathrate hydrate. As underlined by the reviewer, it is reasonable to assume that a similar high level of occupancy is maintained in the sI hydrate of the compressed sample. Nevertheless, the methane occupancy in the newly formed sII clathrate hydrate of the compressed sample could be lower. It is well known that no full occupancy of the small 5^{12} cages is required for the stability of clathrate structures [A. Klapproth et al., *Can. J. Phys.* 81: 503–518 (2003)] and structure II contains twice as many small cages than large cages. If the newly formed sII clathrate hydrate in the sI–sII sample contains 65% occupancy in the small cages and 85% occupancy in the large cages, then approx. 10% of the methane molecules in the sample can be released from the starting sI hydrate during transformation from sI to sII. This fraction of the methane would be available to perform extra-cage translational diffusion in the grain boundary network, together with the fraction of methane released in disordered regions between crystals of sI and sII. Moreover, upon initial formation of cages in the clathrate structure the level of occupancy of the small cages in particular can remain well below the limiting equilibrated case due to the difficulty of filling empty cages after cage formation. This process can take weeks to months due to the low concentration of water vacancies which are needed for this process (see Salamatin et al. 2015, *Energy & Fuels* 29: 5681-5691). Unfortunately, the low-quality diffraction data of our QENS experiment do not allow for establishing the cage fillings with any confidence.

In the revised version of the manuscript, we discuss this issue in Note 2 of the Supplementary Information.

We added the following text to the manuscript (pag.6 and 7):

“Since cage occupancies in the newly formed sII clathrate might be lower compared to the starting sI clathrate hydrate, part of the methane molecules in the starting sI clathrate could indeed have been released from the starting sI hydrate into the grain boundary network during transformation from sI to sII and would be available to perform extra-cage translational diffusion. However, a minimum level of occupancies is required to ensure stability of sII and one can estimate that no more than 10% of the methane in the sample could have been released without destabilization of the water matrix. The existence of a fraction of fast diffusing methane molecules as high as one third strongly suggests that an appreciable fraction of water molecules in the sample are in a disordered state. Such disordered regions would form at the front line of the transformation between clathrate sI and sII, and their sizes are most likely far below the typical size of the crystallites (that is a few micrometers³²). This point is further discussed in Supplementary Note 2.”

To further support our answer, we added to the revised version of the manuscript and the Supplementary Information the results of a QENS study we performed on a methane hydrate sample during transformation from sI to sH at 1.0 GPa. The transformation from sI to sH took several hours at the transition pressure of 1 GPa and this allowed us to measure QENS spectra during coexistence of sI and sH. The quasielastic signal of this sample is very weak, barely visible, compared to the case of sI–sII coexistence (see Supplementary Fig.3). This **strongly supports the peculiarity of the sI–sII clathrate interface** compared to a sample showing a usual temperature-induced or pressure-induced structural transition such as that between sI and sH. Quite clearly, there is much less highly-mobile methane during the sI–sH transition! This result is also consistent with our previous comment on cage occupancies in the new structure compared to the starting structure. Indeed, the water to methane ratio is very low in sH (3.5 according to J. S. Loveday et al., *Nature* 410, 661-663 (2001)) and thus no methane can be possibly released during transformation from sI to sH (even if the small cages of sH are only 50% occupied).

We added the following text to the manuscript (pag.7):

“Moreover, the absence of a prominent quasielastic signal in the spectra of the sI–sH methane clathrate hydrate highlights the very particular nature of the interfaces between coexisting sI and sII, compared to the temperature- or pressure-induced structural transition taking place at high driving forces between two stable forms of methane hydrates such as sI and sH. The micro-structural properties of sI and sII coexisting assemblies certainly deserve to be further investigated.”

2) It remains unclear the possible nature of the polycrystalline material obtained, and hence the nature of the highly mobile methane within this material.

a) There is a lack of any data from secondary measurements to support, for example, the existence of a significant fraction of methane that is “extra-cage”.

This remark is very much related to the previous one (1iii). In our answer to the previous point we explained why the existence of extra-cage methane in the compressed sample can be well understood; in the following, we try to explain why our QENS data provide compelling evidence for that. We also claim that QENS is the most suitable technique to provide such kind of information and that the other techniques applicable to our system would not give the information asked by the reviewer.

First, the measured quasielastic signal is unambiguously associated with dynamics of methane molecules, since methane is the only hydrogenated molecule in the sample. Second, the analysis of the Q dependence of the measured quasielastic signal (pag.7-9 of the manuscript) provides compelling evidence that the probed motion is a random jump translational diffusion. We would like to emphasize that our interpretation of the measured signal as indication of extra-cage fast diffusion of CH₄ does not only result from the time scale of the measured diffusion but also from the analysis of the momentum transfer Q dependence of the signal. As shown in Fig.3 and Supplementary Fig.4, the Q dependence of the signal is well fitted within a Singwi-Sjolander random-jump diffusion model [ref 28 of the manuscript], both in the 3D and the 2D diffusion models. This is **unambiguous indication of an extra-cage diffusion of methane molecules**. No other experimental technique would be able to provide the Q dependence of the motion (in addition to its timescale) as QENS does. Consistently, our analysis provides a small value for the activation energy (0.5 kcal mol⁻¹) which points at van der Waals interactions as main rate-limiting molecular interaction for the observed methane diffusion. As we commented in the new version of the manuscript and in the last

paragraph of our answer to reviewer #1, cage-to-cage hopping of CH₄ in sI clathrates has an activation energy which is more than 1 order of magnitude bigger than our result, as it requires breaking at least one hydrogen bond of the water network [refs 21 and 36 of the manuscript].

The reviewer would welcome secondary measurements to support the evidence of the existence of extra-cage methane. Unfortunately, most experimental techniques cannot be performed at the pressure at which we observe the sI–sII coexistence (0.8 GPa). The techniques that might provide insights into this matter (such as SANS and NMR) are not performed at such high pressures, which are also coupled here with a low temperature study. On the other hand, Raman spectroscopy is commonly performed at high pressure and was performed on sI–sII clathrate samples in the past; however, no interstitial methane could be detected [ref 6 of the manuscript]. Such measurement is complicated by the fact that i) the frequency of the CH stretching for gas methane (2911 cm⁻¹) is very close to those of methane in the small and large cages (2916 and 2904 cm⁻¹, respectively) and that ii) the spatial resolution of a Raman spectrometer is typically 5-10 micrometers in the direction of the beam.

b) There is also limited information on the nature of the material that results from compression. The only other technique (beyond QENS) that was employed to characterize the sample is the powder X-ray diffraction patterns reported in Fig. 1. The presence of the appropriate Bragg peaks do indicate that both sI and sII hydrate structures are present, but otherwise we have no information on the possible nature of the material (for example: Crystallites of the initial sI hydrate are reported to be a few microns in size - does this remain the case? What might be the nature of the grain boundaries?). If it were somehow possible, microscopy images could also be very insightful.

We think that investigating the (micro-)structural details of the polycrystalline sI–sII methane clathrate hydrate is beyond the scope of the present work; the diffraction data obtained during the QENS experiments do not provide a basis for a more detailed analysis. Several experimental and simulation studies have been performed in the past to characterise the nature of the polycrystalline sI–sII methane clathrate hydrate [refs 6-12 of the manuscript]. These studies provide useful hints (see for example ref 6 and our comments above) but no full picture of the phenomenon, and further investigation is certainly needed in the future. Nevertheless, the present work deals with methane dynamics and provides new interesting and unexpected results, regardless of the (micro-)structural details of the polycrystalline material obtained. The observation of a fast translational guest diffusion process in sI–sII methane clathrate hydrates seems to us a sufficient justification for the publication of this paper and we are confident that the present work will stimulate further experiments and simulations on the structural properties of the sI–sII clathrate.

We also would like to emphasize that i) the nature of the obtained polycrystalline material most likely depends on that of the starting sample and ii) even at similar thermodynamic conditions different samples may show different structural properties in cases of metastable states (such as the present one). Hence, any complementary measurement should be performed simultaneously with a QENS measurement to be really conclusive, which is technically not feasible with the state of the art experimental tools. The pattern reported in Fig. 1 is the neutron powder diffraction pattern of the sample and was obtained simultaneously with the QENS data (by considering the elastic scattering only).

Evaluating the crystallite sizes would need synchrotron diffraction data as demonstrated by Chaouachi et al. 2017 (*Crystal Growth & Design* 17: 2458-2472) for Xe- and methane-hydrates at much lower pressures (below 0.4 MPa) and based on our experience we have severe doubts that the sI–sII metastable state could be reproduced easily in a diamond anvil cell and that the data quality from the small amount of sample contained in such a cell would allow for an unambiguous characterisation of crystallite sizes.

Concerning the reviewer's suggestion to perform light-optical or electron microscopy analysis on the sI–sII sample, we are not aware of the possibility to perform electron microscopy under high-pressure conditions. Light-optical microscopy analysis could be performed – yet not simultaneously with the QENS experiment – but micrometer-sized thin layers of sI and sII such as those observed in Fig.3d of ref 6 (and reported above) would be just slightly bigger than the spatial resolution of the technique so very little could be learnt from this study. We intended to corroborate our findings by a real space imaging method, however the available techniques are not applicable to our scientific problem.

Response to comments by reviewer #3:

The authors report enhanced methane diffusion in mixed sI and sII hydrates based on QENS measurement. The measured methane diffusivity ($\sim 10^{-4}$ cm²/s) is 8 orders of magnitude higher than that observed in pure sI hydrates ($\sim 10^{-12}$ cm²/s), and even a factor of 3 to 4 higher than that for pure bulk methane at comparable temperature and pressure. The results are surprising and unexpected and may stimulate great interest in related studies. This work is recommended for publication after minor revision addressing the comments below. Further review is not needed.

We would like to thank the reviewer for recommending the publication of our paper in Nature Communications and for the useful comments. We address all comments below.

1. This is probably the first work that utilizes QENS for determining the methane translational diffusion in clathrate hydrates. While the authors have presented different data analysis to ensure that the measured data correspond to methane translational diffusion, it is somehow strange that a measurement of methane diffusion in pure sI hydrates is not provided. Such data have been obtained by other experimental techniques and can serve as a strong support for the use of QENS for the same purpose. Please provide some reasoning if such a straightforward comparison is not possible.

The reviewer raises an important point. We did measure pure sI methane hydrate by QENS and the spectra do not exhibit any visible quasielastic signal (as we write at the end of the introduction at pag.4 of the manuscript). An example of spectrum is reported in Supplementary Fig.3. This result is fully understood based on the previous studies. Indeed, the diffusion coefficient reported in literature [refs 21 and 24 of the manuscript] for sI at low pressure (of the order of 10^{-11} to 10^{-12} cm²/s) is so low that the corresponding quasielastic signal cannot be observed on any time-of-flight neutron spectrometer.

2. The authors state that the measured diffusivity corresponds to the migration of methane at the interface of two clathrate structures, grain boundary network of sI and sII. This is surprisingly similar to the recent work of Lo et al. (10.1021/jp310972b) where the authors reported enhanced methane diffusion at grain boundary like structures based on molecular dynamics simulations. The authors should reference this work as it is quite relevant.

We thank the reviewer for bringing to our attention the recent work of H. Lo et al. (*J. Phys. Chem. C* 2017, 121, 8280-8289). The latter is now quoted in the manuscript and commented (in red) at pag.4 as follows: "Recently, a study based on molecular dynamics simulations reported diffusion coefficient values in the nanosecond time scale for methane diffusion at grain boundaries-like structures in defected clathrates²⁵." Although very interesting, the results of Lo et al. do not reproduce the diffusion coefficients and the activation energy of our study and thus (as we commented in our answer to reviewer#1) do not provide an explanation for our findings. In particular, their diffusion coefficients are 3 orders of magnitude smaller than in our study.

3. The enhanced methane diffusion in mixed sI-sII seems to imply an increased amount of defects in the crystalline phase compared to pure sI hydrates. However, it has been shown that there can be good interfacing between the two structures through the 5₁₂6₃ cages (see the work of Vatamanu and Peter G. Kusalik (10.1021/ja066515t)). The authors might consider SEM measurements for the morphology for their system (for example, as in the work of Falenty et al. (10.1021/jp310972b)) to better understand their system. If such measurements cannot be done by the authors, it is suggested that they comment on the possible interface structures between the sI and sII and why similar interfaces do not appear in pure polycrystalline sI.

The reviewer is perfectly right. The work of J. Vatamanu and P. G. Kusalik [ref 8 of the manuscript] showed that sII can nucleate upon the surface of sI via an intermediate molecularly-thin layer of 5¹²6³ cages. And if such intermediate layers existed in our sample, they would certainly not allow methane molecules to diffuse

through them as fast as observed in the present work. There is something else going on: The translational diffusion observed in the sI–sII hydrate rather must take place extensively in the grain boundary network or intercalated disordered regions between crystals of sI and sII, as we commented in our answer 1iii to reviewer #2 and in the revised version of the manuscript. The absence of a quasielastic signal in pure clathrate sI can be easily understood as the grain boundary network of sI is not populated by mobile methane molecules. The absence of a prominent quasielastic signal during transformation from sI to sH (see our answer 1iii to reviewer #2 and the revised version of the manuscript) indeed points at the particular nature of sI–sII interfaces, as suggested by the reviewer. A possible explanation is the existence of nanobubbles (mentioned in the old and new version of the manuscript) between crystalline sI and sII regions, possibly arranged in the form of a lamellar structure (shown in Figure 1 of the present document). The peculiarity of the sI–sII clathrate long-lived coexistence is most likely related to the small difference in free energy between sI and sII.

Concerning the use of SEM, please see our answer 2b to reviewer #2. It should be added that i) the rapid quenching procedure used for the work reported in Falenty et al. (10.1021/jp310972b) could not be repeated using the Paris-Edinburgh press employed for the present work and that ii) upon recovery to ambient pressure the microstructures of the sample will almost certainly change.

REVIEWERS' COMMENTS:

Reviewer #1 (Remarks to the Author):

I'm not sure if I'm convinced by the significance of the work, but I'm OK with the recommendation of its publication. I appreciate the QENS experiment is not a trivial one. It still takes a lot of effort to perform such a measurement. What I meant by "routine work" is that I don't see the fundamental scientific value of the result from my point of view. But it is better for the methane and clathrate community, not me, to judge the significance. As to the technical question, I think the authors presented a reasonable interpretation that the highly mobile methane resides near the interfaces, although they don't have solid proof. I might have made similar interpretations as well based on the results. Therefore, I recommend its publication on Nature Communications.

Reviewer #2 (Remarks to the Author):

Having read the reviewer comments and the authors' responses to these, revised manuscript and SI, I am satisfied with the authors' attempts to address my previous comments. I will commend the authors on their thoughtful and well-argued responses to all the reviewer comments, as well as making appropriate revisions to their manuscript (where I believe there is now a convincing case made for their results). I remain intrigued as to what might be going on here, but that is now for future work to uncover. A key question is whether (or not) this behavior is unique to sI/sII hydrate equilibrium, or if it is underpinned by a physical effect with broader implications. I fully support publication in Nature Comm.

Reviewer #3 (Remarks to the Author):

The revised manuscript has address my comments adequately. The higher diffusivity of methane at the sI-sII interface compared to bulk pure methane can be a result of diffusion enhancement in lamellar structure of grain boundary network or the reduced gas density in these structures. The more puzzling issue to me is why such an enhancement is only observed at the interface of sI-sII and not in the grain boundary of pure sI systems. This work brings up many interesting challenges for both experiment and molecular simulations to better understand the interface structure and gas density at the grain boundary of clathrate hydrates.